# Changes in Children’s Adherence to Sustainable Healthy Diets During the Implementation of Chile’s Food Labelling and Advertising Law: A Longitudinal Study (2016–2019)

**DOI:** 10.3390/nu17061041

**Published:** 2025-03-16

**Authors:** Carolina Venegas Hargous, Liliana Orellana, Camila Corvalan, Steven Allender, Colin Bell

**Affiliations:** 1Global Centre for Preventive Health and Nutrition (GLOBE), Institute for Health Transformation, Deakin University, Geelong 3220, Australia; c.venegashargous@deakin.edu.au (C.V.H.); steven.allender@deakin.edu.au (S.A.); 2Biostatistics Unit, Faculty of Health, Deakin University, Geelong 3220, Australia; l.orellana@deakin.edu.au; 3Institute of Nutrition and Food Technology (INTA), University of Chile, Santiago 8330111, Chile; ccorvalan@inta.uchile.cl; 4School of Medicine, Faculty of Health, Deakin University, Geelong 3220, Australia

**Keywords:** obesity, malnutrition, environmental sustainability, global syndemic, triple-duty action, sustainable diet, food policy, food labelling, food marketing restrictions, school-based regulations

## Abstract

**Objectives**: This longitudinal study measured changes in adherence to sustainable healthy diets in 698 Chilean children (aged 3–6 years at baseline) over the period that Chile’s Food Labelling and Advertising Law was implemented. **Methods**: Dietary data were collected annually from 2016 to 2019 applying single multiple-pass 24 h dietary recalls to children’s primary caretakers. The Planetary Health Diet Index for Children and Adolescents (PHDI-C) was used to quantify adherence to sustainable healthy diets where higher scores indicate better adherence. Linear mixed models were fitted to estimate the change in PHDI-C total and individual component scores from 2016 to 2019. **Results**: Mean total PHDI-C score decreased from 50.1 points in 2016 to 46.3 and 46.1 in 2018 and 2019, respectively (*p*-value < 0.001), suggesting that children’s overall adherence to sustainable healthy diets was low and decreased over time. Intake of *legumes*, *fruits*, *dark green vegetables*, *red and orange vegetables*, and *vegetable oils* decreased, while intake of *palm oil*, *red meats*, and *animal fats* increased, resulting in small but significant declines in eight PHDI-C component scores. *Whole cereal* intake increased, while the consumption of *dairy products* and *added sugars* decreased, resulting in improvements in three PHDI-C component scores. **Conclusions**: Aside from the decrease in *added sugar* intake, all dietary changes observed in this study were consistent with trends described among children transitioning from pre-school age to school age. The Law might have contributed to reducing children’s added sugar intake, but further research is required to establish causality.

## 1. Introduction

Worldwide estimates show that 192 million children were affected by moderate and severe underweight in 2016, while 340 million children and adolescents had overweight or obesity [1]. Chile stands out as one of the countries with the highest prevalence of childhood obesity in the world [2]. Although the rate slightly decreased from 24.6% in 2016 to 23.3% in 2023 [3,4], the issue persists alongside several micronutrient deficiencies, including Vitamin D, Vitamin B12, calcium, and iron [5]. All of these forms of malnutrition are expected to worsen as climate change reduces the food system’s capacity to provide healthy, nutritious, and sufficient food for current and future generations [6]. Addressing the common causes of obesity, undernutrition, and climate change promptly and simultaneously requires double-duty and triple-duty actions early in life [7]. Examples of such actions may include policies aimed at improving the quality of food environments by reducing the availability, accessibility, affordability, and desirability of unhealthy foods via taxation, food labelling, media restrictions, and school-based interventions [7]. These policy options have been identified by the Lancet Commission on the Global Syndemic as potential triple-duty actions as they may address obesity by discouraging the consumption of unhealthy packaged foods; undernutrition by encouraging the consumption of healthy minimally processed foods; and climate change by reducing the environmental impact associated with the production, processing, packaging, distribution, storage, preparation, and consumption of unhealthy packaged foods, particularly ultra-processed products [7]. However, evidence of their triple-duty effectiveness is lacking [8,9].

In 2016, Chile implemented a broad suite of national-level policies targeting unhealthy packaged foods with added fats, sugars, or sodium through mandatory front-of-package warning labels, food marketing restrictions, and school-based regulations that ban the sale and/or provision of unhealthy foods in nurseries, primary, and secondary schools [10]. The Food Labelling and Advertising Law (hereafter the Law) is based on a nutrient-profiling model considering energy, saturated fats, total sugars, and sodium with progressively tighter limits implemented in three phases in 26 June 2016 (first phase), 26 June 2018 (second phase), and 26 June 2019 (third phase) [10]. Evidence to date shows that the Law has induced food reformulation to reduce added sugars and sodium in packaged foods and beverages [11,12,13,14]; contributed to reducing household purchases of unhealthy packaged foods and drinks [15,16,17] and of the critical nutrients targeted by the Law (i.e., energy, sodium, sugar, and saturated fats) [18]; and decreased the availability of unhealthy packaged foods in schools [19]. Furthermore, a recent study observed a significant decline in the consumption of total sugars, saturated fats, and sodium in schools among a cohort of 349 Chilean children after three years of the Law being implemented [20]. These changes are in line with the primary aim of the Law of reducing the consumption of unhealthy packaged foods and beverages to address childhood obesity [10]. However, a gap in our knowledge remains as to whether the Law may also perform a triple-duty action, as experts propose [7], by increasing children’s overall adherence to sustainable healthy diets (i.e., diets that promote health and reduce the risk of chronic and foodborne diseases, are nutritionally balanced, are based on a variety of whole or minimally processed plant-based foods, contain moderate amounts of animal-source foods, have a low impact on the environment, and are equitable, accessible, affordable, and culturally adequate [21]). To explore this hypothesis, this study aimed to measure changes in adherence to sustainable healthy diets in a cohort of Chilean children over the period that the Law was implemented.

## 2. Materials and Methods

### 2.1. Study Design

A longitudinal observational study design was used to measure the change in Chilean children’s adherence to sustainable healthy diets from 2016 to 2019. The study is reported following the Strengthening the Reporting of Observational Studies in Epidemiology—Nutritional Epidemiology (STROBE-nut) guidelines [22] (see Appendix A).

### 2.2. Participants and Setting

We used existing data from the Food Environment Chilean Cohort (FECHiC), a study established by the Institute of Nutrition and Food Technology (INTA) to assess the impact of the Law, which are the latest data available on Chilean children’s diets (the only national consumption survey available in Chile was conducted in 2010 [23]). The full FECHiC study design is described in an earlier publication [24]. The sample comprised 961 Chilean children aged 3–6 years from public schools in low–medium income neighbourhoods located in south-east Santiago [24]. The inclusion criteria required children to be singleton births and have no medical conditions that could affect growth or food consumption [24]. Siblings of existing participants were excluded from the study [24]. The baseline sociodemographic and anthropometric characteristics of this sample were similar to those of children from low–medium income neighbourhoods located in urban areas across the country [3,25]. Since 2016, participants have been followed annually through telephone calls and instant messaging to coordinate in-person interviews for data collection purposes, achieving a cohort retention rate of 92.5% (n = 889) in 2017, 83.4% (n = 801) in 2018, and 79.3% (n = 762) in 2019 [20].

### 2.3. Data Collection

Data collection was conducted annually between February and September from 2016 to 2019. The first wave (2016) occurred immediately before the Law was implemented; the second (2017) and third waves (2018) occurred during the first phase of Law implementation, and the fourth wave (2019) occurred during the second phase of the Law implementation (Figure 1).

Trained dietitians collected child and maternal sociodemographic data including age, gender, and level of education via in-person interviews and took direct anthropometric measurements (weight, height, and waist circumference) from children and their mothers following standardised procedures [26]. The children’s weight status (i.e., at risk of undernutrition, normal weight, overweight, obesity, and severe obesity) was determined using the WHO Child Growth Standards [27,28] and maternal weight status (i.e., underweight, normal weight, overweight, and obesity class I-III) was determined using adult body mass index cut-off points [29]. The presence of maternal abdominal obesity was defined as waist circumference >88 cm [30].

Trained dietitians collected the children’s dietary intake information from primary caretakers using single 24 h dietary recalls [24]. They followed the US Department of Agriculture (USDA) automated multiple-pass method [31]. This method helps reduce the risk of recall bias and allows detailed dietary information to be collected [32]. To help participants with portion size estimation, a photographic atlas of Chilean foods and culinary preparations [33] was used. Dietary recall characteristics such as the day of the dietary recall, type of eating pattern, type of diet, and reliability of the dietary recall were also registered.

### 2.4. Merging Nutrient Information Panels and Ingredient Lists with Children’s Dietary Data

Children’s dietary data were merged with an annually updated food composition database [34]. The database was developed for Chile by the University of North Carolina and INTA and included information from the USDA National Nutrient Database [35], which was linked to non-packaged foods such as fruits and vegetables, and from nutrition information panels and ingredient lists of packaged foods available in Chile before and after the implementation of the Law. This information was obtained via the INFORMAS Chile project [36] and involved five waves of data collection conducted annually between January and March 2015–2019 [37] (Figure 1). This enabled the calculation of children’s caloric and macronutrient intake while accounting for the food reformulation that occurred in response to the Law [11,12].

### 2.5. Outcomes of Interest

The Planetary Health Diet Index for Children and Adolescents (PHDI-C) was used to quantify the children’s adherence to sustainable healthy diets [38]. The PHDI-C was adapted from the Planetary Health Diet Index developed and validated by Cacau et al. [39] to better account for children’s nutritional requirements [38]. The PHDI-C consists of 16 components organised into foods that should be promoted (adequacy components), foods that should be consumed in a balanced manner (ratio and optimum components), and foods that should be limited (moderation components). Each component is associated with a recommended range of percentage of total caloric intake and a continuous scoring scale that adds to a total PHDI-C score of 150 points [38].

The PHDI-C component scores were calculated using the decision tree and food disaggregation methodology developed by Venegas Hargous et al. [38]. Briefly, calories from whole foods and culinary ingredients reported in children’s dietary recalls were directly allocated into the corresponding index component (e.g., calories from apples were directly allocated into the *Fruits* PHDI-C component). Calories from composite foods were first disaggregated into main energy sources and then allocated into the corresponding index components (e.g., calories from sweet biscuits were disaggregated into calories from cereals, added sugars, and palm oil (if the main source of fat declared in the ingredient list was palm oil), and then these calories were allocated into the *Cereals*, *Added sugars*, and *Palm oil* PHDI-C components, respectively). We used foods’ nutrient information panels and ingredient lists to guide this process.

Once the calorie allocation process was finalised, we calculated the percentage of calories consumed from each index component relative to the total non-alcoholic caloric intake reported by each child. The only exceptions were ratio components, for which we calculated the percentage of calories consumed from dark green vegetables and red and orange vegetables relative to total calories consumed from all vegetables, and the percentage of calories consumed from whole cereals relative to total calories consumed from all cereals (refined and whole). These percentages were then compared to the recommended percentages of total caloric intake defined for each PHDI-C component and scores were calculated using the formulae provided by Venegas Hargous et al. [38].

The scoring system works as follows: Zero points are assigned to consumption levels falling outside of the recommended range of percentage of total caloric intake [38]. For adequacy components, the highest scores are awarded to consumptions meeting or exceeding the recommended levels, while lower scores are assigned proportionally to those below the recommendation [38]. For the ratio and optimum components, higher scores are assigned to consumptions that align closely with the recommendation, while proportional reductions are applied to those either above or below the recommendation [38]. For moderation components, consumption levels closer to zero are assigned higher scores, whereas values above zero are scored proportionally lower [38]. Higher scores reflect higher adherence to sustainable healthy diets [38].

The outcomes of interest were (1) percentage of total caloric intake from each PHDI-C component, (2) PHDI-C individual component scores, and (3) total PHDI-C score.

### 2.6. Covariates

The covariates included in the analysis were day of dietary recall (weekday vs. weekend/holiday); type of eating pattern on the day of the recall (typical vs. atypical (due to a celebration, sickness, and/or a holiday)); type of diet on the day of the recall (normal (i.e., diet with no dietary restrictions) vs. special (i.e., vegan, vegetarian, lactose-free, or gluten-free)); child gender (female or male); age (3–4 years or 5–6 years); weight status (non-overweight, overweight, or obese); and maternal age (<25 years or ≥25 years); and education (incomplete secondary education, complete secondary education, or complete tertiary education), which was used as a proxy for socioeconomic position [40]. These variables were selected as covariates because they showed significant associations with at least one PHDI-C component in a cross-sectional analysis of FECHiC participants’ baseline data (2016) [41].

### 2.7. Statistical Analysis

Participants with missing dietary data (n = 3) and those who missed at least one wave of data collection (n = 245) were excluded from this study. Given that no data collection occurred after July 2019 (Figure 1), all participants who completed data collection after July in previous study years (i.e., 2016–2018) were excluded to maintain consistency (n = 15). The analytical sample included 698 participants aged 3–6 years at baseline who completed the four years of data collection (Appendix A).

Linear mixed models were fitted to estimate the mean for each outcome of interest (i.e., percentage of total caloric intake from each PHDI-C component, PHDI-C individual component scores, and total PHDI-C score) at each year of data collection, and the mean changes from baseline (2016) to the first (2017 and 2018) and second (2019) phases of the Law’s implementation. The models included, as fixed effects, the year of data collection (categorical variable) and the set of covariates previously mentioned to account for potential confounding; as a random effect, the child unique identifier was used to account for the longitudinal structure of the data. Estimates alongside 95% confidence intervals (CI) are reported for each outcome.

A sensitivity analysis considering participants who had data collected at baseline (2016) and at least one more wave was performed (n = 877). All statistical analyses were conducted in Stata v17. Statistical significance was defined as a *p*-value < 0.01 to account for multiple testing.

## 3. Results

### 3.1. Participants’ Baseline Characteristics (Table 1)

At baseline (2016), half of the participants were female and more than 70% were 3–4 years of age. Most of the children’s mothers were ≥ 25 years of age (n = 575, 82%), and had completed secondary education (n = 404, 58%). Moreover, 51% percent of children had a normal weight status, 29% were overweight, and 17% were obese. Seventy-one percent (n = 476) of the children’s mothers were affected by overweight or obesity, and more than half (n = 366, 55%) had abdominal obesity. According to the children’s mothers, most dietary recalls recorded information from a weekday (n = 598, 86%) and reflected children’s typical dietary pattern (n = 586, 84%) and children’s normal diet (n = 661, 95%). INTA’s dietitians classified 94% (n = 654) of the dietary recalls as reliable. The baseline characteristics were similar among the excluded participants, apart from child gender and maternal level of education, with excluded participants consisting of a higher proportion of females and children whose mothers had not completed secondary education (Appendix A). See Appendix A for participants’ characteristics across follow-up years, including the change in the nutritional composition of the children’s diets.

**Table 1 nutrients-17-01041-t001:** Participants’ baseline characteristics.

	Analytical Sample (n = 698)
**Sociodemographic characteristics**	**n (%)**
**Child gender**	
Male	351 (50.3)
Female	347 (49.7)
**Child age**	
3–4 years	505 (72.4)
5–6 years	193 (27.6)
**Maternal age**	
<25 years	123 (17.6)
≥25 years	575 (82.4)
**Maternal level of education**	
Incomplete secondary education	108 (15.4)
Complete secondary education	404 (57.9)
Complete tertiary education	186 (26.7)
**Anthropometric characteristics**	**n (%)**
**Child weight status ^a^**	
At risk of undernutrition	21 (3.0)
Normal weight	357 (51.2)
Overweight	204 (29.2)
Obesity	77 (11.0)
Severe obesity	39 (5.6)
**Maternal weight status ^b,c^**	
Underweight	5 (0.8)
Normal weight	188 (28.1)
Overweight	241 (36.0)
Obesity class I	155 (23.2)
Obesity class II	53 (7.9)
Obesity class III	27 (4.0)
**Maternal abdominal obesity ^c,d^**	
Absence	303 (45.3)
Presence	366 (54.7)
**Dietary recall characteristics**	**n (%)**
Day of the dietary recall	
Weekday	598 (85.7)
Weekend day/holiday	100 (14.3)
**Type of eating pattern on the day of the dietary recall ^e^**	
Typical	586 (84.0)
Atypical	112 (16.0)
**Type of diet on the day of the dietary recall ^f^**	
Normal	661 (94.7)
Special	37 (5.3)
**Reliability of the dietary recall ^g^**	
Reliable	654 (93.7)
Unreliable	44 (6.3)

^a^ Defined based on WHO Child Growth Standards 2006 [27] for children under 5 years old and WHO Growth Reference 2007 [28] for children above 5 years of age. ^b^ Defined based on the WHO cut-off points for BMI in adults [29]. ^c^ A total of 29 participants included in the analytical sample had missing data, 28 were pregnant, and 1 refused to be measured. ^d^ Defined based on the Adult Treatment Panel III criteria for the clinical identification of metabolic syndrome (waist circumference > 88 cm) [30]. ^e^ Typical eating pattern refers to a dietary recall from a regular day; atypical eating pattern refers to a dietary recall from a special occasion (e.g., celebration, vacation, or sickness). ^f^ Normal diet refers to a diet with no dietary restriction of any kind; special diet refers to a diet with any dietary restriction (e.g., vegan, vegetarian, lactose-free, or gluten-free). ^g^ Unreliable recalls are those with missing information on the amount consumed of any type of food.

### 3.2. Changes in Children’s Dietary Intake and PHDI-C Scores

Adjusted estimates of children’s percentage of total caloric intake from the 16 PHDI-C components across study years and the changes relative to baseline are presented in Table 2. Adjusted estimates of children’s PHDI-C components scores and total PHDI-C score across study years and the changes relative to baseline are presented in Table 3. Non-adjusted estimates are presented in Appendix A, respectively.

The children’s mean total PHDI-C score was low in 2016 (50.1 [95%CI 49.08, 51.15]) and decreased significantly in 2018 (−3.79 [95%CI −5.14, −2.45]) and 2019 (−4.06 [95%CI −5.40, −2.72]) compared to 2016, suggesting that the children’s adherence to sustainable healthy diets decreased over the period the Law was implemented (*p* < 0.001). Small but significant reductions were observed for eight of the sixteen PHDI-C components scores, while significant improvements were observed in three component scores.

#### 3.2.1. Changes in Adequacy Components

The percentage of total caloric intake from nuts and peanuts was very low and showed no statistically significant changes over time. The percentage of total caloric intake from *legumes* was lower than recommended and decreased in 2018 and 2019 compared to 2016, resulting in statistically significant mean score reductions for the *legumes* PHDI-C component from 2016 to 2019 (−0.71 [95%CI −1.00, −0.42]). The percentage of total caloric intake from *fruits* was also lower than recommended and decreased over time (although not statistically significantly), resulting in a statistically significant mean score reduction for the *fruits* PHDI-C component from 2016 to 2019 (−0.67 [95%CI −1.06, −0.28]). The change in the percentage of total caloric intake from the *vegetables* PHDI-C component was small and inconsistent over time, resulting only in a statistically significant mean score reduction for the *vegetables* PHDI-C component in 2018 compared to 2016 (−0.51 [95%CI −0.82, −0.2]).

#### 3.2.2. Changes in Ratio Components

The percentage of calories from *dark green vegetables* and *red and orange vegetables* was lower than recommended and decreased slightly over time (not statistically significant). This resulted in small but statistically significant mean score reductions for the *dark green vegetables* and *red and orange vegetables ratio PHDI-C* components in 2018 (−0.14 [95%CI −0.23, −0.04] and −0.48 [95%CI −0.67, −0.29], respectively) and 2019 (−0.11 [95%CI −0.21, −0.02] and −0.30 [95%CI −0.49, −0.12], respectively) compared to 2016. In contrast, the percentage of total caloric intake from *whole cereals* increased from 2016 to 2019, resulting in statistically significant mean score improvements for the *whole cereals ratio* PHDI-C component in 2017 (0.27 [95%CI 0.10, 0.45]) and 2019 (0.52 [95%CI 0.34, 0.69]) compared to 2016.

#### 3.2.3. Changes in Optimum Components

The percentage of total caloric intake from cereals increased over time, becoming closer to the recommended intake; however, the mean scores for the cereals PHDI-C component showed no significant change from 2016 to 2019. The percentage of total caloric intake from tubers and potatoes was above the recommendation and showed no statistically significant changes over time. The percentage of total caloric intake from dairy products was above the recommendation but decreased over time, resulting in statistically significant mean score improvements for the dairy products PHDI-C component in 2018 (0.54 [95%CI 0.20, 0.88]) and 2019 (0.46 [95%CI 0.11, 0.80]) compared to 2016. The percentage of total caloric intake from eggs and white meats was close to the recommendation and showed no statistically significant changes over time. Small but non-significant reductions in the percentage of total caloric intake from vegetable oils were observed across follow-up years, shifting slightly away from the recommended intake. This resulted in statistically significant mean score reductions for the vegetable oils PHDI-C component in 2018 (−0.41 [95%CI −0.69, −0.12]) and 2019 (−0.56 [95%CI −0.85, −0.27]) compared to 2016.

#### 3.2.4. Changes in Moderation Components

The percentages of total caloric intake from *palm oil* and *animal fats* were above the recommendation and increased over time, resulting in significant mean score declines in the *palm oil* and *animal fats* PHDI-C components in 2017 (−0.54 [95%CI −0.98, −0.09] and −0.68 [95%CI −1.16, −0.19], respectively), 2018 (−1.16 [95%CI −1.6, −0.71] and −0.91 [95%CI −1.4, −0.43], respectively), and 2019 (−1.27 [95%CI −1.72, −0.83] and −0.97 [95%CI −1.45, −0.49], respectively) compared to 2016. A similar trend was observed for *red meat* consumption, with mean scores significantly decreasing in 2018 (−0.69 [95%CI −1.19, −0.2]) and 2019 (−0.73 [95%CI −1.22, −0.24]) compared to 2016. In contrast, the percentage of total caloric intake from *added sugars* significantly decreased from 2016 to 2017 and remained lower over 2018 and 2019. This resulted in significant mean score improvements in the *added sugars* PHDI-C component in 2017 (0.25 [95%CI 0.10, 0.40]), 2018 (0.23 [95%CI 0.08, 0.38]), and 2019 (0.32 [95%CI 0.17, 0.47]) compared to 2016.

#### 3.2.5. Sensitivity Analysis

Sensitivity analysis considering participants with dietary data collected at baseline and at least one more follow-up year produced similar estimates (Appendix A). The only exception was the mean score for tubers and potatoes, which showed significant declines from 2016 to 2019 (Appendix A).

## 4. Discussion

There was a significant decline in adherence to sustainable healthy diets in a cohort of Chilean children over the period when Chile’s Food Labelling and Advertising Law was implemented, particularly during 2018 and 2019 compared to 2016. Notably, this was observed over the same period during which children transitioned from preschool age (3–6 years) to school age (6–9 years) and most of the dietary changes observed were in line with declines in diet quality observed in other studies over the same life-stage [42,43,44]. Counter to these trends was the decline in children’s added sugar intake [44,45,46], particularly in 2017 and sustained over 2018 and 2019. This dietary change may be partly explained by the significant reduction in the content of total sugars in packaged food and beverages observed after each phase of the Law’s implementation [11,14]. However, the absence of a control group in this natural experiment prevents us from attributing this dietary change solely to the implementation of the Law. Since age-related changes in diet quality may account for several of this study’s observations, further research is required to establish causality.

The hypothesis for the current study was that the Law may contribute to increasing Chilean children’s adherence to sustainable healthy diets by discouraging the consumption of unhealthy and unsustainable packaged foods with a high content of energy, saturated fats, added sugar, and/or sodium, while encouraging the consumption of healthy and sustainable foods, particularly minimally processed plant-based foods, as their replacement [7]. The findings from this study likely support the first part of this hypothesis by showing significant improvements in the mean score for the added sugars PHDI-C component from 2016 to 2019, which resulted from a significant decline in children’s added sugar intake from 2016 to 2017, which was sustained over 2018 and 2019 (from 16.3% [95%CI 15.7%, 16.9%] of total caloric intake in 2016 to 13.5% [95%CI 12.9%, 14.0%] in 2017, 13.9% [95%CI 13.3%, 14.4%] in 2018 and 13.9% [95%CI 13.4%, 14.5%] in 2019). This dietary change contrasts with observations from a longitudinal study conducted by Pinto da Costa et al. [44] among Brazilian children (n = 5013) showing a significant increase in the percentage of children consuming ≥ 1 serving per day of sweet foods (52.5% to 65.6%, *p* < 0.001) and soft drinks (23.8% to 34.3%, *p* < 0.001) as they transitioned from 4 to 7 years of age. It is also contrary to observations from Welsh et al. [45] and Neri et al. [46] on data from the National Health and Nutrition Examination Survey (NHANES, 2009–2014) on children living in the US (n = 8136 and n = 9469, respectively) showing a significant increase in the contribution of added sugars to children’s total caloric intake as they transition from 2–5 to 6–11 years of age (from 12.7% [95%CI 12.4%, 13.1%] to 14.9% [95%CI 14.5, 15.3%] and from 12.1% ± 0.2% to 15.3 ± 0.2%, respectively). The decline in added sugar intake observed in our study is consistent with findings by Fretes et al. [20] who reported a significant decline in total sugar intake among a sample of 349 FECHiC participants after the first and second phases of the Law’s implementation. Since the Law targets all packaged foods and beverages in which sugars, saturated fats, and/or sodium have been added and exceed the limits established by the nutrient profiling model [10], it is possible that the decline in added sugar intake observed in this and Fretes et al.’s studies is partly explained by the Law’s implementation. Reyes et al. [11] noted a significant reduction in the content of total sugar among several packaged foods and beverages, including breakfast cereals, yogurts, milk and milk drinks, sweet spreads, and soft drinks, after the first phase of the Law implementation. Concurrently, Zancheta et al. [13] observed a significant increase in the use of non-nutritive sweeteners in yogurts, milk drinks, desserts and ice creams, and beverages after the first phase of the Law implementation. Moreover, Rebolledo et al. [14] observed further reductions in the content of total sugars in breakfast cereals, yogurts, milk and milk drinks, sweet spreads, desserts and ice creams, processed fruits, and beverages after the second and third phases of the Law implementation. Likely as a consequence of these changes in the food supply, Smith-Taillie et al. [16,18] observed a significant decrease in the purchases of products high in sugar after the first and second phases of the Law implementation compared to a counterfactual scenario based on a 36-month pre-policy timeframe. Notably, Rebolledo et al. observed a significant increase in the purchases of non-sugar-sweetened beverages from 2016 to 2017 [47] and a significant increase in the consumption of non-nutritive sweeteners among 875 FECHiC participants after the first phase of the Law implementation (from 78% in 2016 to 92% in 2017) [34]. This evidence suggests that this cohort of Chilean children may have partially replaced the consumption of added sugars with non-nutritive sweeteners during the Law’s implementation. However, further research is required to establish causation.

Regarding the second part of this study’s hypothesis suggesting that the Law may help increase the consumption of sustainable healthy foods, particularly minimally processed plant-based foods, our findings showed small declines in the intake of *legumes*, *fruits*, *dark green vegetables*, *red and orange vegetables*, and *vegetable oils*, and small increases in the intake of *palm oil*, *animal fats*, and *red meats*. This resulted in a small but significant decline in children’s total PHDI-C score in 2018 and 2019 compared to 2016, which was not offset by the improvements in three PHDI-C components scores due to the increased intake of *whole cereals* and decreased intake of *dairy products* and *added sugars*.

Aside from the decrease in *added sugar* intake, the dietary changes observed in this study are consistent with descriptions from longitudinal studies in children of similar age (3–9 years) showing declines in diet quality as children transition from pre-school age to school age [42,43,44]. Rauber et al. [42] observed a significant decrease in the proportion of Brazilian children meeting the Healthy Eating Index dietary recommendations for *fruit*, *milk*, *grains*, *saturated fats*, *meat, and legumes* from 3–4 to 7–8 years of age (n = 345 and 307, respectively). Pinto da Costa et al. [44] observed a decrease in adherence to the WHO’s dietary recommendations among a cohort of Portuguese children (n = 5013) from 4 to 7 years of age, with significant reductions in the consumption of *fruits*, *vegetables*, and *dairy products*, and significant increases in the consumption of *meat*, *fish*, *eggs*, *sweet foods*, and *soft drinks*. Lastly, Mannino et al. [43] observed a significant decrease in the percentage of non-Hispanic white girls meeting the US Food Guide Pyramid recommendations for *fruits*, *vegetables*, and *dairy products* from 5 to 9 years of age (n = 181). Also in line with our findings, Mannino et al. [43] observed a significant increase in dietary fibre intake, which is consistent with findings from cross-sectional studies showing higher intakes of *whole grains*, particularly from breakfast cereals, among school-aged children compared to pre-school-aged children [48,49].

Notably, minimally processed foods and processed culinary ingredients containing naturally occurring levels of saturated fats, sugars, and sodium (e.g., nuts and seeds, fruits, vegetables, legumes, red and white meats, table sugar, table salt, and cooking oils) were not regulated by the Law [10]. Neither were processed and ultra-processed foods in which nutrients of concern had been added but fell below the limits established by the Law (e.g., dairy products, breakfast cereals, artificially sweetened beverages) [10]. Furthermore, the Law did not regulate the sale of unhealthy foods outside of schools and the prevalence of unhealthy foods within 100 m from the schools remained unacceptably high after full Law implementation [50]. The high availability of unhealthy foods outside of schools [50], together with the evidence showing that children tend to increase their daily energy intake from foods bought outside of school as they become older [51], might help explain the decline in adherence to sustainable healthy diets in this cohort of Chilean children in 2018 and 2019 compared to 2016. It is possible that children’s response to the Law restricting the availability of unhealthy foods inside of schools was to source these foods outside of school. This hypothesis is supported by the findings by Fretes et al. [20] showing that FECHiC participants decreased their intake of total sugars, saturated fats, and sodium inside of schools, but increased the consumption of these nutrients at other locations (i.e., restaurants, streets, and public transportation) after the first and second phases of the Law implementation. In particular, the overall increase in saturated fat intake described by Fretes et al. [20] aligns with our findings showing increased palm oil and animal fat intake from 2016 to 2019. These results are consistent with the observed increases in saturated fat intake as children grow [42], and may be explained by the high availability of unhealthy foods outside of schools [50] and the non-significant reduction in the content of saturated fats in most packaged foods after the Law’s implementation [11,12,14]

### Strengths and Limitations

The strengths of this study include the longitudinal study design and the availability of high-quality data before and after each phase of the Law implementation. This enabled us to estimate the change in adherence to sustainable healthy diets in a cohort of Chilean children as a way of exploring whether the Law may perform as a triple-duty action by increasing children’s adherence to healthy, nutritionally adequate, and environmentally sustainable diets. Trained dietitians followed the USDA five-step multi-pass method [31] to obtain children’s dietary data through a single 24 h dietary recall applied to children’s primary caretakers. This likely helped reduce the risk of recall bias [32]. Additionally, the use of a photographic atlas of Chilean foods and culinary preparations [33] and a bespoke food composition database, including information from packaged foods and beverage available in Chile before and after the Law implementation [34], enabled us to accurately estimate portion sizes and calculate energy and macronutrient intakes considering food composition changes occurring after the Law’s implementation [11,12]. Finally, the use of the PHDI-C enabled us to quantify adherence to sustainable healthy diets as a proxy indicator of triple-duty outcomes while taking into consideration children’s specific nutritional needs [38]. Although the PHDI-C is not yet validated, it is based on a previously validated index that has been positively associated with overall diet quality and negatively associated with carbon emissions [39]. Further research is required to assess the potential implications of the small but significant dietary changes observed in this study in terms of health and environmental outcomes.

The internal validity of this study was limited by the absence of a control group given the national implementation of the Law. This constrained our ability to attribute observed changes to the sole effect of the Law and determine whether the decrease in children’s adherence to sustainable healthy diets would have been different had the Law not been implemented (counterfactual). This limitation could have been addressed if we had access to dietary data from a parallel cohort of similar children not exposed to the Law or four years of data collection prior to the Law implementation so the counterfactual could have been estimated. However, none of these alternatives were available for this study.

Another limitation affecting external validity is that FECHiC participants are a convenience sample of Chilean children from low–middle income neighbourhoods of Santiago, Chile, and, hence, are not representative of the entire Chilean population. That said, they had similar baseline characteristics to children from low–medium income neighbourhoods located in urban areas across the country [3,25]. In addition, the use of single 24 h recalls prevents us from providing a representative measure of children’s usual dietary intake [52]; however, most dietary recalls were reported as children’s typical intake, and models included dietary recall characteristics such as the day of the dietary recall, the type of eating pattern, the type of diet, and the reliability of the dietary recall as potential confounders. Plus, a single 24 h dietary recall is still considered a valid method for estimating a population’s mean dietary intake [53]. Another limitation of this type of dietary recall is the risk of recall bias [53], which was reduced to some extent by (1) having trained dietitians conducting the 24 h recalls following the USDA multiple-pass method [32] during face-to-face interviews with children and their mothers, and (2) asking the child whenever the primary caretaker was unsure of what was eaten during school hours. Nevertheless, there is still a risk of misreporting, particularly of foods eaten away from home. Conducting three 24 h dietary recalls over non-consecutive days including weekdays and weekends would have been the most accurate method for obtaining children’s dietary intake [52]; however, these data were not available for this cohort of Chilean children. Finally, while the PHDI-C provides an indication of children’s adherence to sustainable healthy diets, it does not measure global syndemic impacts directly. Long-term longitudinal studies are needed to understand the effects of the Law on obesity, undernutrition, and climate change associated with reducing the production, processing, packaging, distribution, storage, preparation, and consumption of unhealthy packaged foods, particularly ultra-processed foods. Furthermore, detailed food-related environmental impact databases will be required in order to conduct such analyses.

## 5. Conclusions

This study is the first to explore whether Chile’s Food Labelling and Advertising Law performed a triple-duty action. All dietary changes observed in this study, except for the decline in *added sugar* intake, were consistent with descriptions from longitudinal studies among children transitioning from pre-school age to school age [42,43,44]. This suggests that the Law may have contributed to reducing added sugar intake, but may not have offset other declines in diet quality and associated declines in environmental sustainability as the children aged. However, further research involving counterfactual models in new jurisdictions will be required to examine causality and whether these types of policies can help improve children’s overall adherence to sustainable healthy diets.

These findings add to the existing literature highlighting the need for a comprehensive set of complementary triple-duty actions in addition to the Law to improve Chilean children’s adherence to sustainable healthy diets [20]. Evidence to date points to integrating sustainability into food-based dietary guidelines, improving the healthiness and reducing the environmental impact of school meal programmes, and implementing multi-component interventions that increase children’s exposure to healthy and environmentally sustainable foods [8]. Actions in Chile may include using the recently updated sustainable dietary guidelines for the Chilean population [54,55] as an overarching policy to improve the school feeding programme by increasing the provision of legumes, nuts and peanuts, fruits, and vegetables while decreasing the provision of red meats and ultra-processed foods. Establishing a purchasing criterion in favour of minimally processed plant-based foods that restricts the acquisition of unhealthy packaged foods by schools is also an option. Lastly, implementing multi-component interventions in schools, including vegetable gardens and cooking classes focused on increasing children’s exposure to healthy and environmentally sustainable foods, may also contribute to increasing their adherence to sustainable healthy diets. Future studies should focus on identifying which of these actions are most suitable in the Chilean context as well as piloting and evaluating their potential triple-duty benefits.

## Figures and Tables

**Figure 1 nutrients-17-01041-f001:**
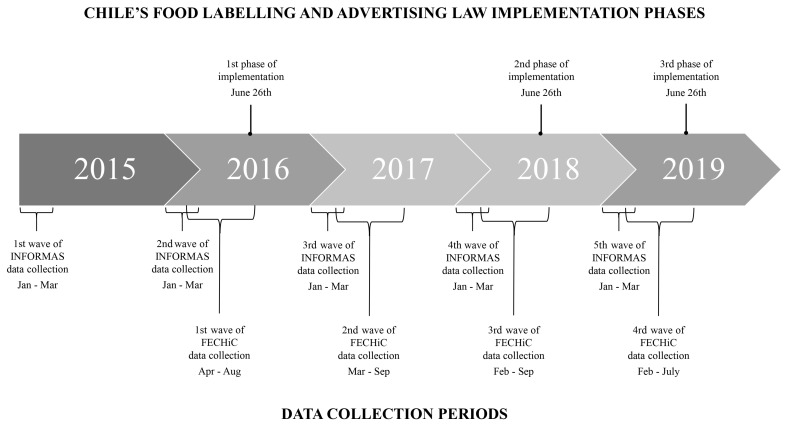
Chile’s Food Labelling and Advertising Law implementation phases and data collection periods for the Food Environment Chilean Cohort (FECHiC) study and INFORMAS project.

**Table 2 nutrients-17-01041-t002:** Adjusted changes in children’s percentage of total caloric intake from PHDI-C components after Law implementation (n = 698).

PHDI-C Components	PHDI-C Recommended Percentage of Total Caloric Intake for Children ^a^	Children’s Percentage of Total Caloric Intake ^a,b^									
		2016	2017	2018	2019	2017 vs. 2016		**2018 vs. 2016**		**2019 vs. 2016**	
	Optimal Value (Range)	Mean (95%CI)	Mean (95%CI)	Mean (95%CI)	Mean (95%CI)	Diff (95%CI)	*p*-Value	**Diff (95%CI)**	** *p* ** **-Value**	**Diff (95%CI)**	** *p* ** **-Value**
**Adequacy components**											
Nuts and peanuts	≥11.6 (0, 100)	0.34 (0.19, 0.49)	0.39 (0.24, 0.54)	0.44 (0.29, 0.59)	0.27 (0.12, 0.42)	0.05 (−0.16, 0.25)	0.654	0.1 (−0.11, 0.3)	0.349	−0.07 (−0.28, 0.13)	0.489
Legumes	≥11.3 (0, 100)	2.23 (1.89, 2.57)	2.05 (1.71, 2.39)	1.56 (1.22, 1.9)	1.06 (0.72, 1.40)	−0.18 (−0.66, 0.29)	0.456	−0.67 (−1.14, −0.20)	0.005	−1.18 (−1.65, −0.7)	<0.001
Fruits	≥5.0 (0, 100)	4.93 (4.55, 5.32)	4.8 (4.41, 5.19)	4.5 (4.11, 4.89)	4.49 (4.10, 4.88)	−0.14 (−0.64, 0.37)	0.596	−0.43 (−0.93, 0.07)	0.091	−0.44 (−0.94, 0.06)	0.083
Vegetables	≥3.1 (0, 100)	1.43 (1.32, 1.54)	1.46 (1.35, 1.57)	1.28 (1.17, 1.39)	1.48 (1.37, 1.59)	0.03 (−0.12, 0.17)	0.715	−0.15 (−0.29, −0.01)	0.041	0.05 (−0.09, 0.19)	0.490
**Ratio components**											
DGV ratio	29.5 (0, 100)	5.66 (4.4, 6.91)	4.99 (3.73, 6.26)	4.62 (3.36, 5.88)	4.49 (3.24, 5.75)	−0.66 (−2.41, 1.09)	0.46	−1.03 (−2.78, 0.71)	0.246	−1.16 (−2.9, 0.58)	0.191
ReV ratio	38.5 (0, 100)	40.25 (37.86, 42.64)	39.81 (37.41, 42.22)	36.77 (34.38, 39.16)	38.58 (36.19, 40.96)	−0.44 (−3.71, 2.84)	0.794	−3.48 (−6.74, −0.23)	0.036	−1.67 (−4.92, 1.58)	0.313
WC ratio	75.0 (0, 100)	4.89 (3.74, 6.04)	7.62 (6.46, 8.77)	6.19 (5.04, 7.33)	9.3 (8.16, 10.45)	2.73 (1.18, 4.27)	0.001	1.3 (−0.24, 2.84)	0.098	4.41 (2.88, 5.95)	<0.001
**Optimum components**											
Cereals	30.0 (0, 60.0)	25.32 (24.57, 26.08)	27.09 (26.33, 27.85)	28.69 (27.94, 29.45)	29.44 (28.69, 30.20)	1.76 (0.74, 2.78)	0.001	3.37 (2.36, 4.38)	<0.001	4.12 (3.11, 5.13)	<0.001
Tubers and potatoes	1.6 (0, 3.1)	3.4 (3, 3.79)	3.71 (3.31, 4.11)	3.72 (3.33, 4.12)	3.68 (3.28, 4.07)	0.31 (−0.24, 0.87)	0.272	0.33 (−0.23, 0.88)	0.249	0.28 (−0.27, 0.83)	0.318
Dairy products	12.2 (0, 24.4)	19.95 (19.17, 20.73)	18.75 (17.97, 19.53)	16.49 (15.71, 17.27)	16.07 (15.29, 16.84)	−1.2 (−2.16, −0.24)	0.014	−3.46 (−4.42, −2.51)	<0.001	−3.89 (−4.84, −2.93)	<0.001
Eggs and white meats	6.2 (0, 12.2)	5.74 (5.2, 6.29)	6.01 (5.46, 6.56)	6.62 (6.07, 7.16)	6.61 (6.07, 7.15)	0.27 (−0.47, 1.01)	0.474	0.87 (0.14, 1.61)	0.019	0.87 (0.14, 1.60)	0.020
Vegetable oils	14.1 (0, 28.3)	10.41 (9.91, 10.92)	10.31 (9.80, 10.82)	9.83 (9.32, 10.33)	10.00 (9.49, 10.50)	−0.1 (−0.79, 0.59)	0.770	−0.59 (−1.27, 0.1)	0.094	−0.41 (−1.1, 0.27)	0.236
**Moderation components**											
Palm oil	0.0 (0, 2.4)	3.37 (3.02, 3.72)	3.78 (3.42, 4.13)	4.5 (4.15, 4.85)	4.67 (4.32, 5.02)	0.4 (−0.07, 0.87)	0.094	1.13 (0.66, 1.60)	<0.001	1.30 (0.83, 1.77)	<0.001
Red meats	0.0 (0, 2.4)	4.43 (3.89, 4.97)	5.33 (4.79, 5.87)	5.78 (5.24, 6.32)	5.62 (5.08, 6.16)	0.9 (0.17, 1.63)	0.016	1.35 (0.63, 2.08)	<0.001	1.19 (0.47, 1.92)	0.001
Animal fats	0.0 (0, 1.4)	1.77 (1.53, 2.01)	2.22 (1.98, 2.46)	2.41 (2.17, 2.65)	2.41 (2.17, 2.66)	0.45 (0.12, 0.79)	0.008	0.64 (0.31, 0.98)	<0.001	0.64 (0.31, 0.97)	<0.001
Added sugars	0.0 (0, 4.8)	16.3 (15.73, 16.87)	13.46 (12.89, 14.03)	13.87 (13.3, 14.43)	13.92 (13.36, 14.49)	−2.84 (−3.58, −2.1)	<0.001	−2.43 (−3.17, −1.7)	<0.001	−2.38 (−3.12, −1.64)	<0.001

Abbreviations: PHDI-C, Planetary Health Diet Index for children and adolescents; DGV ratio, dark green vegetables ratio; ReV ratio, red and orange vegetables ratio; WC ratio, whole cereals ratio; CI, confidence interval; Diff, difference. ^a^ All values are expressed as percentage of total caloric intake. The only exceptions are the values for the DGV ratio and ReV ratio components, which are expressed as percentage of total calories from all vegetables, and the values for the WC ratio component, which are expressed as percentage of total calories from all cereals (refined and whole). ^b^ Estimates and *p*-values from mixed-effects models adjusting for dietary recall characteristics including day of the dietary recall (weekday vs. weekend/holiday), type of eating pattern (typical (i.e., recall from a typical day) vs. atypical (i.e., recall from a special occasion such as celebrations, sickness, or vacations)), and type of diet (normal (i.e., a diet with no dietary restriction of any kind) vs. special diet (e.g., vegan, vegetarian, lactose free, or gluten free)), and child and maternal baseline characteristics, including child gender (male vs. female), child age (3–4 years vs. 5–6 years), child weight status (non-overweight, overweight, and obesity), maternal age (<25 years vs. ≥25 years), and maternal education (incomplete secondary education, complete secondary education, complete tertiary education); n = 698.

**Table 3 nutrients-17-01041-t003:** Adjusted changes in children’s PHDI-C total and individual component scores after Law implementation (n = 698).

PHDI-C Components	PHDI-C Possible Scores	Participants’ PHDI-C Scores ^a^									
		2016	2017	2018	2019	2017 vs. 2016		2018 vs. 2016		2019 vs. 2016	
	Points	Mean (95%CI)	Mean (95%CI)	Mean (95%CI)	Mean (95%CI)	Diff (95%CI)	*p*-Value	Diff (95%CI)	*p*-Value	Diff (95%CI)	*p*-Value
**Adequacy components**											
Nuts and peanuts	0–10	0.23 (0.13, 0.32)	0.29 (0.19, 0.39)	0.34 (0.24, 0.44)	0.21 (0.11, 0.31)	0.07 (−0.07, 0.2)	0.327	0.11 (−0.02, 0.25)	0.102	−0.01 (−0.15, 0.12)	0.845
Legumes	0–10	1.46 (1.25, 1.67)	1.42 (1.21, 1.63)	1.09 (0.88, 1.3)	0.75 (0.54, 0.96)	−0.04 (−0.33, 0.25)	0.803	−0.37 (−0.66, −0.08)	0.013	−0.71 (−1.00, −0.42)	<0.001
Fruits	0–10	5.97 (5.67, 6.27)	5.72 (5.42, 6.03)	5.6 (5.29, 5.9)	5.30 (5.00, 5.60)	−0.25 (−0.64, 0.15)	0.220	−0.37 (−0.76, 0.02)	0.064	−0.67 (−1.06, −0.28)	0.001
Vegetables	0–10	4.2 (3.96, 4.44)	4.07 (3.82, 4.31)	3.69 (3.45, 3.94)	4.11 (3.87, 4.35)	−0.14 (−0.45, 0.17)	0.388	−0.51 (−0.82, −0.2)	0.001	−0.09 (−0.4, 0.22)	0.566
**Ratio components**											
DGV ratio	0–5	0.37 (0.29, 0.44)	0.34 (0.27, 0.42)	0.23 (0.16, 0.3)	0.25 (0.18, 0.32)	−0.02 (−0.12, 0.08)	0.663	−0.14 (−0.23, −0.04)	0.006	−0.11 (−0.21, −0.02)	0.024
ReV ratio	0–5	2.37 (2.23, 2.5)	2.19 (2.05, 2.32)	1.89 (1.75, 2.02)	2.07 (1.93, 2.2)	−0.18 (−0.37, 0.01)	0.057	−0.48 (−0.67, −0.29)	<0.001	−0.3 (−0.49, −0.12)	0.001
WC ratio	0–10	0.59 (0.46, 0.72)	0.86 (0.73, 0.99)	0.81 (0.67, 0.94)	1.1 (0.97, 1.23)	0.27 (0.1, 0.45)	0.002	0.22 (0.04, 0.39)	0.015	0.52 (0.34, 0.69)	<0.001
**Optimum components**											
Cereals	0–10	7.05 (6.9, 7.21)	7.22 (7.06, 7.38)	7.17 (7.01, 7.32)	7.18 (7.03, 7.34)	0.17 (−0.05, 0.39)	0.136	0.11 (−0.11, 0.33)	0.318	0.13 (−0.09, 0.35)	0.250
Tubers and potatoes	0–10	0.46 (0.35, 0.57)	0.31 (0.2, 0.43)	0.35 (0.24, 0.47)	0.32 (0.21, 0.43)	−0.14 (−0.3, 0.01)	0.071	−0.1 (−0.26, 0.05)	0.192	−0.14 (−0.3, 0.01)	0.074
Dairy products	0–10	3.77 (3.52, 4.03)	4.12 (3.86, 4.37)	4.31 (4.06, 4.57)	4.23 (3.97, 4.48)	0.34 (−0.00, 0.69)	0.051	0.54 (0.2, 0.88)	0.002	0.46 (0.11, 0.80)	0.009
Eggs and white meats	0–10	2.87 (2.62, 3.13)	2.92 (2.67, 3.18)	3 (2.75, 3.26)	2.96 (2.70, 3.22)	0.05 (−0.31, 0.42)	0.773	0.13 (−0.23, 0.49)	0.477	0.09 (−0.27, 0.45)	0.629
Vegetable oils	0–10	5.57 (5.36, 5.78)	5.33 (5.12, 5.54)	5.17 (4.96, 5.38)	5.01 (4.80, 5.22)	−0.24 (−0.53, 0.04)	0.097	−0.41 (−0.69, −0.12)	0.006	−0.56 (−0.85, −0.27)	<0.001
**Moderation components**											
Palm oil	0–10	4.77 (4.44, 5.1)	4.23 (3.9, 4.56)	3.61 (3.28, 3.94)	3.49 (3.17, 3.82)	−0.54 (−0.98, −0.09)	0.019	−1.16 (−1.6, −0.71)	<0.001	−1.27 (−1.72, −0.83)	<0.001
Red meats	0–10	4.89 (4.53, 5.24)	4.47 (4.11, 4.83)	4.19 (3.83, 4.55)	4.16 (3.8, 4.52)	−0.42 (−0.91, 0.07)	0.095	−0.69 (−1.19, −0.2)	0.006	−0.73 (−1.22, −0.24)	0.004
Animal fats	0–10	5.38 (5.03, 5.74)	4.71 (4.35, 5.06)	4.47 (4.12, 4.82)	4.41 (4.06, 4.76)	−0.68 (−1.16, −0.19)	0.006	−0.91 (−1.4, −0.43)	<0.001	−0.97 (−1.45, −0.49)	<0.001
Added sugars	0–10	0.17 (0.06, 0.28)	0.42 (0.31, 0.53)	0.4 (0.29, 0.52)	0.49 (0.38, 0.60)	0.25 (0.1, 0.4)	0.001	0.23 (0.08, 0.38)	0.002	0.32 (0.17, 0.47)	<0.001
**Total PHDI-C score**	0–150	50.11 (49.08, 51.15)	48.64 (47.6, 49.68)	46.32 (45.28, 47.35)	46.05 (45.01, 47.08)	−1.47 (−2.83, −0.12)	0.033	−3.79 (−5.14, −2.45)	<0.001	−4.06 (−5.4, −2.72)	<0.001

Abbreviations: PHDI-C, Planetary Health Diet Index for children and adolescents; DGV ratio, dark green vegetables ratio; ReV ratio, red and orange vegetables ratio; WC ratio, whole cereals ratio; CI, confidence interval; Diff, difference. ^a^ Estimates and *p*-values from mixed-effects models adjusting for child and maternal baseline characteristics, including child gender (male vs. female), child age (3–4 years vs. 5–6 years), child weight status (non-overweight, overweight, and obesity), maternal age (<25 years vs. ≥25 years), and maternal education (incomplete secondary education, complete secondary education, complete tertiary education), and dietary recall characteristics including day of the dietary recall (weekday vs. weekend/holiday), type of eating pattern (typical (i.e., recall from a typical day) vs. atypical (i.e., recall from a special occasion such as celebrations, sickness, or vacations)), and type of diet (normal (i.e., a diet with no dietary restriction of any kind) vs. special diet (e.g., vegan, vegetarian, lactose free, or gluten free)); n = 698.

## Data Availability

The de-identified data described in the manuscript, code book, and analytic code will not be made available publicly due to the ongoing nature of the cohort study, but can be made available on reasonable request. Proposals should be directed to the corresponding author, who will then pass the proposal on to members of the Center for Research in Food Environments and Prevention of Nutrition-related Chronic Diseases (CIAPEC)—INTA for deliberation and approval. To gain access, data requestors will need to sign a data access and collaboration agreement.

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
