# Peer review of "Changes in Children’s Adherence to Sustainable Healthy Diets During the Implementation of Chile’s Food Labelling and Advertising Law: A Longitudinal Study (2016–2019)"

_nutrients, 2025, doi:10.3390/nu17061041_

Round 1
Reviewer 1 Report
Comments and Suggestions for Authors
Interesting article adding to the evidence of effectiveness of the Chilean Law. Overall well written. What is missing for me in the context of this study is the trend in prevalence in overweight and obesity in adults and children in Chile since the implementation of the Law. I would expect to see this in the introduction and discussion. What has happened with the prevalence of overweight and obesity and why?
Another comment is related to the dietary index score PHDI-C. It needs more explanation on how it was calculated. See also specific comments.
A few questions and unclarities remain. See below for specific comments.
Abstract
L18 dietary recalls of primary care-takers
L26 Question: How can you specify with a dietary index score how much palm oil is consumed? Is the score based on ingredient lists on the label?
Introduction
L39-55: I miss the prevalence of overweight/obesity and undernutrition in Chile and also what has happened since the implementation of the Law.
L78: aka diets: What does it mean?
Methods
L148-165: The explanation of how the PHDI-C was calculated could be supported by a figure, either in the text or in the supplementary materials. It is not entirely clear to me how the researchers arrived at e.g. the estimates for palm oil and animal fats consumption. On the basis of which data?
L181-183: Unclear sentence: ‘no data collection occurred after July 2029’… and in the second part of the sentence ‘participants who completed data collection after July’ which suggest that there was data collection after July.
Results
Table 1. Questions:
Were there siblings included in the study? If yes How many? What do the classes of obesity mean, please add more explanation to the legend.
L242-252: How were these ratio components calculated? See comment above and below.
Discussion
L418 Since the PHDI-C is not validated, it is even more important to clarify how it is calculated. Especially how researchers arrived at estimations of palm oil and animal fats.
L420 Palm oil is a difficult ingredient. In terms of land use (the amount of oil per hectare) it is the most efficient. While at the same time there are great environmental issues with it.
Author Response
Dear Reviewer, we really appreciate your feedback on this manuscript. Please find attached a point-by-point response to your comments.

Reviewer 2 Report
Comments and Suggestions for Authors
The paper “Changes in children’s adherence to sustainable healthy diets over the period Chile’s Food Labelling and Advertising Law was implemented: a longitudinal study (2016-2019)” measured changes in adherence to sustainable healthy diets among a cohort of 698 Chilean children (aged 3-6 years at baseline) over the period Chile’s Food Labelling and Advertising Law was implemented. The research content is quite interesting. Here are some specific issues.
Comments:
Q1. Why did this article choose 698 Chilean children (baseline age 3-6 years old) as volunteers? Is it because there is a limitation in analyzing the relationship between diet and health for this age group?
Q2. How is the accuracy of dietary data ensured?
Q3. The volunteers vary from year to year. Is it possible to select those who have completed the data collection throughout the process for the result analysis?
Q4. The study attributed the decrease in added sugars to the law, but did not rule out other factors (such as media campaigns, changes in family health awareness). It is suggested to further enhance the relevant expressions.
Q5. The sample was drawn from low-income neighborhoods in San Diego and may not be representative of rural or high-income groups.
Q6. Additional details of Ethics committee approval are recommended.
Q7. The conclusions of the study (such as reduction of added sugars and decline in overall diet quality) are partially repeated with the existing literature, and it is recommended to emphasize the innovation of the "triple responsibility" perspective, and combine environmental indicators to highlight the contribution.
Q8. The conclusions indicate that the law may reduce added sugars, but the recommendations do not specify how to further verify this cause-and-effect relationship. Additional research directions are recommended.
Author Response

(The authors gave the same response as above.)
